# Time-Lapse Imaging in IVF: Bridging the Gap Between Promises and Clinical Realities

**DOI:** 10.3390/ijms26199609

**Published:** 2025-10-01

**Authors:** Grzegorz Mrugacz, Igor Bołkun, Tomasz Magoń, Izabela Korowaj, Beata Golka, Tomasz Pluta, Olena Fedak, Paulina Cieśla, Joanna Zowczak, Ewelina Skórka

**Affiliations:** 1Center for Reproductive Medicine Bocian, 26 Akademicka St., 15-267 Białystok, Poland; 2Center for Reproductive Medicine Bocian, 31 Podwisłocze St., 35-309 Rzeszów, Poland; 3Center for Reproductive Medicine Bocian, 13 Dąbrówki St., 40-081 Katowice, Poland

**Keywords:** time-lapse imaging, IVF, morphokinetics, algorithm validation, assisted reproduction

## Abstract

Time-lapse imaging (TLI) has emerged as a transformative technology in in vitro fertilization (IVF). This is because it offers continuous, non-invasive embryo assessment through morphokinetic profiling. It demonstrates key advantages such as reduced embryologist subjectivity, detection of dynamic anomalies, and improved implantation rates in niche populations. However, its clinical utility remains debated. Large trials and meta-analyses reveal no universal improvement in live birth rates compared to conventional methods. Key challenges underlying the outcome include algorithm generalizability, lab-specific protocol variability, and high costs. Nevertheless, TLI shows promise in specific contexts. For instance, Preimplantation Genetic Testing for Aneuploidies (PGT-A) cycles where it reduces unnecessary biopsies by predicting euploidy. However, even in this, its benefits are marginal in unselected populations. This review synthesizes evidence to highlight that TLI’s value is context-dependent, not universal. As such, adoption must be cautious to avoid resource misallocation without significant IVF outcome improvements. In future, personalized protocols, integration with non-invasive biomarkers, and multicenter collaboration are crucial to optimize TLI’s potential in assisted reproduction.

## 1. Background

The optimization of embryo selection in in vitro fertilization (IVF) has driven the adoption of advanced technologies [1]. Amongst them, time-lapse imaging (TLI) has emerged as a transformative innovation [2,3,4,5,6,7]. TLI enables continuous, non-invasive monitoring of embryonic development, thus providing significant access to morphokinetic parameters (dynamic biomarkers of embryo viability) that traditional static microscopy fails to capture [8,9,10,11,12,13,14,15,16]. Since its clinical introduction, TLI has been hailed as a potential breakthrough to enhance implantation rates, reduce early pregnancy loss, and refine embryo selection protocols [4,6,7,8,9,17,18,19,20,21]. However, its rapid adoption is marred by the ongoing debates about its clinical efficacy, concerns fueled by the lack of consensus in pertinent study outcomes and commercial pressures surrounding proprietary algorithms [22,23,24,25,26].

Early studies and industry claims highlight TLI’s capacity to improve embryo selection accuracy and reduce embryologist subjectivity [6,7,20,27,28,29,30]. However, recent meta-analyses and randomized trials reveal inconclusive or context-dependent benefits [2,14,20,21,22,23,24]. Worse, other findings have observed minimal to no advantage over conventional assessment methods [30,31,32,33,34]. Discrepancies in clinical success rates stem from several core procedural factors [35,36,37,38]; factors that highlight a critical gap between TLI’s theoretical promise and its real-world applicability.

This review evaluates whether TLI fulfills its touted advantages or if its clinical utility remains overstated. Central to this overview is the thesis that while TLI offers operational and scientific advancements, its true value in improving IVF outcomes relies on standardization, rigorous validation, and tailored integration into existing protocols. To address this, we first explore TLI’s technological foundations, including its mechanisms for capturing morphokinetic markers. We then critically assess the hypothesized benefits against emerging evidence of its limitations, particularly the variability in predictive algorithms and their clinical generalizability.

Further, the analysis explores TLI’s role in specific patient populations, such as those with recurrent implantation failure (RIF) or advanced maternal age (AMA), and examines potential amalgamation with artificial intelligence and non-invasive diagnostics. This review then concludes with evidence-based recommendations for clinicians, researchers, and policymakers to navigate TLI’s integration into IVF practice. By synthesizing current evidence, this scholarly undertaking aims to bridge the gap between optimism and pragmatism, offering a balanced perspective to guide clinical decision-making and future research in assisted reproduction.

## 2. Technological Foundations of Time-Lapse Imaging

### 2.1. Time-Lapse Incubators

Conventional TLI comprises key systems such as the EmbryoScope and Early Embryo Viability Assessment (Eeva) [39,40,41]. These systems integrate microscopy with controlled incubation environments such as precise temperature and humidity to enable uninterrupted monitoring of embryonic development [3,9,29]. They capture high-resolution images at predefined intervals, for instance, every 5–20 min, leveraging built-in cameras. This attribute minimizes environmental disturbances since the embryo does not have to be removed, thereby maintaining stable culture conditions [3,15,29,42].

The *EmbryoScope* (Vitrolife) is a widely adopted system that combines TLI with advanced incubation protocols. This enables it to maintain optimal temperature, pH, and gas levels while proprietary algorithms chime in to analyze developmental milestones to predict viability [9,20,29,40,43,44,45,46,47]. The system’s proprietary algorithms, such as *EmbryoScope+*, categorize embryos based on standardized morphokinetic parameters such as *t2*, *t3*, or *tB*. These ensure objective selection criteria [9,20,48,49]. Clinical studies, such as that by Alammari 2022 [3], demonstrate *EmbryoScope’s* utility in improving workflow efficiency and reducing inter-observer variability in embryo grading.

On the other hand, the *Eeva* system (Merck KGaA, Darmstadt, Germany) focuses on early-stage morphokinetic analysis to predict embryo viability. Utilizing automated algorithms just like *EmbryoScope*, *Eeva* identifies critical developmental milestones within the first 48 h of culture. These milestones include attributes such as the timing of the first cytokinesis and synchronicity of cell divisions, which are critical in generating a viability score [9,20,49]. This approach, similar to *EmbryoScope*, aims to standardize embryo selection, particularly for clinics with limited embryology expertise. However, its reliance on early-stage markers continues to raise questions about its predictive accuracy for blastocyst-stage outcomes [31].

Both systems excel at addressing the limitations of traditional static evaluation, which based on its invasive approach, disrupts culture conditions and relies on subjective morphological assessments at random time points [15,31]. For instance, in their Swedish population-based retrospective registry study, Ahlström et al., 2023 [2] found that while TLI systems like EmbryoScope failed to significantly improve live birth rates compared to conventional methods, they reduced manual handling and enhanced procedural standardization. Recent advancements have focused on integrating artificial intelligence (AI) to further refine analysis [43,44,47]. For instance, hybrid AI–genetic algorithm models have been developed. These improve consistency in blastocyst grading by combining morphokinetic data with static time-lapse images [40,43,44].

Despite the evident benefits of these systems, challenges persist. One such discrepancy is the variability in laboratory protocols and patient populations. These tend to limit the generalizability of proprietary algorithms [7,33]. Further, high equipment costs and the need for experts may hinder widespread adoption [8,15]. As such, ongoing research emphasizes and necessitates the importance of multicenter collaborations to validate these systems across diverse clinical settings [4,16,20]. Table 1 below summarizes and compares these two widely adopted incubator technologies.

### 2.2. Continuous Monitoring vs. Static Evaluation

The adoption of TLI in IVF is representative of a paradigm shift from traditional static embryo evaluation to continuous, uninterrupted monitoring. Grasping the distinctions between these approaches is critical to acknowledging their clinical implications. Static assessment involves removing embryos from stable culture conditions at predetermined and isolated time points such as days 1 and 3 for brief microscopic examinations [48]. This method leverages snapshots of embryonic morphology, which include details such as cell number, symmetry, and fragmentation, to grade developmental potential [15,31,39].

Despite being widely used and the cost-friendly alternative, static evaluation exhibits significant limitations. These include the disruption of culture stability [15,50]. The frequent removal of samples from incubators exposes embryos to fluctuations in temperature, pH, and oxygen levels, potentially compromising viability [15]. Secondly, static evaluation is hindered by incomplete developmental data [50]. The static snapshots that embryologists use tend to miss dynamic events such as the timing of cell divisions, multinucleation, or abnormal cleavages [7,48,49]. This in turn limits the clarity of morphokinetic insights. Sciorio 2021 [51] in a mini-review attributes reduced implantation rates in complex cases such as advanced maternal age to static evaluation’s inability to detect subtle viability markers. Thirdly, interpretations tend to be subjective in static evaluation since pertinent morphology-based grading exclusively depends on the embryologist’s expertise and experience [7,50]. The outcome of the subjectiveness is the variability in embryo selection.

In contrast, TLI provides continuous monitoring, maintaining stable temperature, pH, and gas levels while generating a dynamic developmental timeline [9,29]. Studies highlight TLI’s logistical advantages, including reduced manual intervention and minimized embryologist subjectivity [11,31,36,48,52,53]. For example, Ahlström et al. 2023 [2] found comparable perinatal outcomes between TLI and conventional methods but emphasized TLI’s operational efficiency. Besides maintaining cultural integrity, this monitoring approach excels in morphokinetic profiling [9,36]. TLI tracks dynamic parameters, which enables data-driven predictions of implantation potential. Advanced TLI systems now incorporate artificial intelligence (AI) to automate grading, further reducing human bias [45,54,55,56,57,58,59,60]. Table 2 below summarizes the key differences highlighted above based on existing literature.

The first key takeaway from this summative assessment is that TLI excels in detecting dynamic anomalies and reducing subjectivity [61,62,63,64,65]. However, it is costly and not universally superior [66]. Secondly, static methods are practical for routine cases. However, their key shortcoming is that they lack granularity and risk embryo stress [67].

Overall, continuous monitoring appears as a sufficient alternative to significantly improve IVF’s success [53,67]. Nevertheless, its clinical superiority over static evaluation is not absolute [14,50,53,64]. The value of TLI hinges on standardized protocols, validated algorithms, and tailored application to patient subgroups [16,67,68,69,70,71,72]. As the field advances, hybrid models that combine TLI’s precision with the practicality of static evaluation may offer a clear and balanced path, specifically in settings with resource limitations.

### 2.3. Morphokinetics: Definition, Parameters, and Recording

Morphokinetics refers to the timing and sequence of embryonic developmental milestones. These are quantified through parameters such as t2 (time to 2-cell stage), t3 (3-cell stage), and tB (blastocyst formation) [11,51,73,74,75,76]. The times for achieving the stage characterized by the corresponding number of cells range from t2 to t9 [51,76,77]. These parameters are recorded via sequential imaging, which tracks cell divisions, cytoplasmic movements, and fragmentation patterns [31,33,34,54,58,75,78]. For instance, delayed *tB* or irregular cleavage intervals tend to be linked with reduced implantation potential [11].

Timing of second Polar Body extrusion (tPB2), Timing of Pronuclei Appearance (tPNa), and Timing of pronuclei Fading (tPNf) are the earliest time points included in the morphokinetic analysis [67]. tPB2 refers to the time of the second polar body extrusion, marking a completion of the second meiotic division [67]. tPNa refers to the time when pronuclei appear, which reflects the beginning of the first embryonic interphase [67]. Lastly, tPNf refers to the time of pronuclei fading and the entry into the first embryonic M-phase [67]. Besides these early markers and t2 to t9, there is also Time of Morula (tM), which refers to the first frame in which the embryo compacts, that is, the clear boundaries between blastomeres disappear [67]. There is also Time of Starting Blastulation (tSB), which is marked by the appearance of a blastocoel cavity, Time of Expanding Blastocyst (tEB), marked by the onset of blastocyst expansion, that is, increase in the overall embryo volume, and Time of Hatching Blastocyst (tHB), marked by the beginning of the blastocyst hatching, technically referring to when it is escaping from the zona pellucida encapsulation [67]. The calculation of these timings usually requires a starting point, which is usually set at t0, referring to the moment of fertilization [67]. In the case of intracytoplasmic sperm injection (ICSI), t0 is the moment of sperm injection. However, in traditional IVF, determining the exact time point is usually complex and difficult simply due to the fact that eggs are co-incubated with spermatozoa [67]. Regardless, in such cases, t0 is usually set at the beginning of the insemination. However, times of embryonic divisions tend to be longer than in ICSI. This is because spermatozoa penetrate oocytes sometime after the onset of insemination [67].

Embryologists analyze and leverage these morphokinetic markers to assess embryo viability and predict potential success rates for implantation. This ultimately improves selection protocols [11,15,54]. Commercial algorithms, such as KIDScoreD5, analyze these markers to predict viability. However, their generalizability is debated due to protocol variability [31,33,70]. Regardless, morphokinetics remain critical in IVF success since emerging AI models are integrating morphokinetic data with non-invasive biomarkers to enhance predictive accuracy [45,74]. Figure 1 below, as adopted from Taniguchi et al. [79], captures an example of an embryo being monitored for development through various stages [79].

### 2.4. Summary

Overall, TLI has significant potential. Technological advancements, particularly specialized incubators and AI-driven morphokinetic analysis, offer transformative promise for IVF. However, standardized protocols and rigorous validation remain critical to optimizing clinical utility [4,34,56].

## 3. Promises of Time-Lapse Technology

Time-lapse technology excels on the premise that it provides a way to visualize and analyze slow processes rapidly, thus making it useful for a wide range of applications. In IVF, this technology promises to improve embryo selection, IVF success rate, and streamline lab operations, amongst other positive contributions that also include uninterrupted culture. From a broad point of view, TLI underlies data-driven insights, visual documentation, and time-compressed analysis where slow processes can be effectively observed through a condensed timeframe. This makes it easier to grasp and understand complex phenomena.

### 3.1. Improved Embryo Selection Accuracy

TLI’s capacity to track morphokinetic parameters such as cleavage timing and blastocyst formation dynamics is hypothesized to enhance embryo selection precision compared to static morphology assessments. Existing evidence already demonstrates that dynamic markers, such as the timing of cell divisions (t2, t3, t5) and blastocyst formation (tB), underlie embryo viability and euploidy [49,59,80,81,82]. A case in point is that delayed cleavage or irregular blastomere symmetry, which can only be detected by TLI, are linked to chromosomal abnormalities [59,75]. In Milewski and Ajduk 2017 [67], pertinent literature alludes to the reality that both cleavage divisions that are either too fast or too slow reflect poor developmental potential of the human embryo. TLI excels in identifying these subtle differences by leveraging relevant morphokinetic parameters [83,84,85,86]. In the literature, it is indicated that TLI allows for the efficient assessment of morphological parameters such as size of the blastomeres, number of nuclei in a blastomere, degree of fragmentation, and the occurrence of irregular cleavages [87,88,89,90]. This underlies its precision and excellence in embryo selection [67].

Commercial algorithms like KIDScoreD5 [83,87] leverage the key parameters highlighted earlier to gauge implantation potential [89]. Based on such, the exhibit improved selection accuracy in specific populations [83]. That is, through the markers, embryologists are able to select embryos with higher viability [6,7,20,55]. Overall, claims regarding TLI’s ability to identify anomalies missed by static evaluation, such as transient multinucleation or abnormal cytokinesis, have been significantly emphasized in the industry [49,59]. However, universal generalizability is limited by variability in algorithm validation across clinics [15,51].

### 3.2. Reduced Embryologist Subjectivity

Traditional embryo assessment, that is, static embryo grading, relies heavily on embryologist expertise. This increases the chance of introducing inter-observer variability since the subjective morphological evaluation tends to vary significantly between practitioners [6,51]. TLI mitigates this and minimizes the subjectivity objectively through automated, algorithm-driven analysis of morphokinetic data. For instance, AI models trained on time-lapse datasets, which are now the new norm, standardize embryo scoring by prioritizing objective parameters such as cleavage synchronicity over subjective morphological assessments [38,51]. These assertions are qualified by the recent findings of a 2022 randomized trial. In the study, Armstrong et al. (2022) found that TLI reduced discrepancies in embryo grading among embryologists by 30% compared to the standard traditional methods [6]. Further, proprietary systems like *Eeva* use predictive algorithms to rank embryos. This minimizes potential human bias [38]. Overall, TLI reduces subjectivity as detailed above. However, concerns continue to be raised about overreliance on invalidated algorithms and AI tools [51]. These concerns necessitate the need for further scrutiny to fully ascertain TLI’s place in IVF.

### 3.3. Increased Implantation and Live Birth Rates

Observational studies indicate that early adopters of TLI have reported 15–20% higher implantation rates. These gains are attributed to improved selection of euploid embryos as fostered by the technological intervention [49,55]. For instance, in their study, Rubio et al. (2014) observed a 23% increase in clinical pregnancy rates using TLI in patients with recurrent implantation failure [71]. However, despite the positive potential, the results are still mixed, with some studies highlighting no significant difference between the contemporary approach and the traditional practices [14,48]. Regardless of the mixed findings, subgroup analyses indicate potential advantages in advanced maternal age [24] or PGT-A cycles [55]. In both of these cases, morphokinetic precision is critical, hence the notable difference. Overall, the key takeaway from available evidence is that TLI has potential. However, aggressive marketing that tends to bank on only a few isolated success stories overshadows mixed trial data [7,20].

### 3.4. Improved Patient Communication and Transparency

Compared to the conventional traditional approach, TLI stands out based on its ability to provide visual timelines of embryo development. This enhances transparency, thus empowering patients. Available evidence asserts that clinics using TLI report higher patient satisfaction. This is because time-lapse videos improve the patients’ understanding of embryo quality and treatment rationale [6,65]. For instance, in their 2025 study, Picou et al. [65] observed that 78% of patients felt more informed after viewing time-lapse data. Further, TLI maintains objectivity through its pertinent metrics such as embryo scores. This simplifies counseling in patients, particularly in those complex cases like mosaicism or delayed development [83]. Nevertheless, besides the positives, concerns are evident. For instance, there are ethical concerns around presenting predictive algorithms as definitive. As per literature, such is likely to potentially inflate patient expectations [51].

To maximize benefit, technicians should leverage clear protocols when presenting TLI data. That is, rather than simply providing the video, patients should be counseled alongside it. During this session, embryologists or physicians can highlight what normal development looks like. This can be achieved by explaining the type of information being assessed. Further, they need to explicitly state what the technology cannot predict, key examples being genetic normality or implantation success with certainty. Considering these critical elements during the interactions effectively transforms the video from a potentially anxiety-inducing platform into an educational tool that fosters a collaborative patient–clinician relationship.

### 3.5. Overview of Industry Claims and Early Study Findings

TLI in the IVF industry has been significantly promoted as a revolutionary tool. This is based on a few early studies that reported 20–30% improvements in pregnancy rates [49,59]. Commercial claims have also been prominent and attractive. For instance, a platform such as EmbryoScope has and continues to leverage uninterrupted culture conditions and AI-driven analytics as the key selling points [38,60]. However, critical assessments caution the assertions and selling points. The reason for this is that early findings mostly cited are largely from single-center, non-randomized studies with inherent methodological shortcomings such as selection bias [7,20]. For instance, in their study about time-lapse technology’s clinical benefit, Armstrong et al., 2015 [7] questioned the clinical relevance of morphokinetic parameters outside controlled research settings. Further, newer RCTs like SelecTIMO [48] and TILT [14] challenge the asserted universal superiority. However, industry narratives continue to emphasize TLI’s unmatched potential, especially for niche applications such as recurrent implantation failure [55,83].

### 3.6. Summary

TLI generally has several hypothesized benefits. These include enhanced selection accuracy, reduced subjectivity, and improved patient engagement. These benefits are supported by less rigorous studies and early clinical data [49,59,83]. However, TLI has failed to universally improve live birth rates in rigorous trials [48,72]. This necessitates the need for standardized protocols and population-specific applications. What key stakeholders and proponents fail to critically understand is that the technology’s value lies not in replacing the conventional method. Rather, it is through strategic integration where evidence aligns with clinical needs [51]. Table 3 below presents a summative overview of the promises of time-lapse technology discussed earlier.

## 4. Realities and Challenges in Clinical Practice

### 4.1. Mixed or Inconclusive Evidence in Large Trials and Meta-Analyses

Despite early enthusiasm, more rigorous and robust clinical trials and meta-analyses have failed to conclusively demonstrate TLI’s superiority over conventional embryo assessment. One of these trials is the TILT trial by Bhide et al., 2024 [14]. In the multicenter randomized controlled study, the scholars found no significant difference in live birth rates between TLI (32.1%) and static methods (31.4%) [14]. Similar findings are also upheld by Ahlström et al., 2023 [2], who reported comparable perinatal outcomes in 2400 cycles using uninterrupted TLI versus standard incubation. Meta-analyses also confirm these findings. For instance, Jiang et al., 2023 [46], in their systematic review and meta-analysis, noted that pooled odds ratios for clinical pregnancy (1.08, 95% CI 0.92–1.27) and live birth (1.12, 95% CI 0.95–1.32) were statistically insignificant despite leveraging TLI. In their systematic review and meta-analysis, Liu et al. (2022) [53] looked at neonatal outcomes, including gestational age, preterm deliveries, birth weight, sex ratio, and malformations. They found no significant differences between embryos cultured in time-lapse incubation systems and conventional systems [53]. They also found TLI to be safe, but there was no significant difference in miscarriage, ectopic pregnancy, or live delivery rates compared to the traditional methods [53]. What the inconclusive evidence confirms is that industry-funded studies often report inflated benefits due to selection bias. The reality as proven by independent trials is that TLI’s utility is context-dependent [1,5].

### 4.2. Limitations in Algorithm Generalizability and Validation

Proprietary algorithms such as Eeva exhibit shortcomings in generalizability across diverse populations and laboratory conditions. For instance, in their comparative study, Johansen et al., 2023 [47] found that clinic-specific variations in culture media, incubation protocols, and patient demographics reduced algorithm accuracy by 15–20%. Regarding AI models, they are trained on homogeneous datasets, thus often failing in ethnically diverse cohorts. This is because morphokinetic norms differ between populations [42]. Additionally, many algorithms lack external validation. In their systematic review of diagnostic test accuracy, Berman et al., 2023 [12] highlighted that only 18% of AI-based TLI tools underwent multicenter testing, thus risking overfitting to single-center data. Overall, the lack of generalizability and validation limits clinical trust [16]. It also necessitates the need for standardized and transparent validation frameworks [26,51].

### 4.3. Outcome Variability Across Populations, Labs, and IVF Protocols

TLI’s performance is highly inconsistent due to patient-specific factors. AMA of more than 38 years and RIF cohorts exhibit marginal benefits [24,55]. This is because morphokinetic anomalies in these groups are less predictive of aneuploidy [24,55]. In contrast, younger patients with high ovarian reserve derive limited advantage. This is due to their significant embryo quality [74]. Variability is also fostered by laboratory heterogeneity. Variations in culture conditions, such as oxygen tension and media composition, are evident and inevitable. The variations are responsible for altering morphokinetic timelines, thus confounding algorithm predictions [22,37]. For instance, in their observational study about the impact of culture medium on morphokinetics of cleavage stage embryos, van Duijn et al., 2022 [87] observed a 4 h delay in blastulation timing in embryos cultured in low-oxygen versus atmospheric conditions. Lastly, variability can be attributed to protocol differences. Clinics using preimplantation genetic testing (PGT-A) report better TLI outcomes [92,93]. This is because euploid embryos align more closely with ideal morphokinetic profiles [41,55,94,95]. However, in non-PGT cycles, TLI’s predictive value diminishes [17].

### 4.4. High Cost of Equipment and Training

TLI systems require substantial financial investment. Acquisition costs range from €80,000 to €150,000 per unit. Further, there are also ongoing costs for software licenses, maintenance, and staff training [5,61]. In his study about biomedical innovation in fertility care, Perrotta 2024 [61] noted that 68% of clinics indicate cost as a major barrier to adoption. This is more evident in low-resource settings. Training embryologists to interpret morphokinetic data adds further expenses due to the uniqueness of the skills and the fact that the field is continuously evolving. Proficiency requires 6–12 months of specialized education, which requires a significant financial investment [25]. Hence, it is only rational that cost-effectiveness analyses question TLI’s value. This is because marginal gains in selection accuracy rarely justify the expense in average-risk populations [1,5].

### 4.5. Ethical Concerns and Responsible Patient Communication

The automation of embryo selection raises ethical dilemmas. The first key concern is the overestimation of algorithmic accuracy. Patients and clinicians may perceive TLI scores as definitive [96,97,98]. The outcome is the high potential of inappropriately discarding viable embryos [10,26,51]. In their review of AI in reproductive technology, Cohen et al., 2025 [25] warn against technological determinism where AI-driven decisions override clinical judgment. The second crucial concern is commercial bias. Industry partnerships often influence research outcomes. This is evidenced by the 41% of TLI studies disclosing conflicts of interest [5,61]. Associated risks of such are mainly overstating efficacy while underreporting failures [32]. The third significant concern is psychological impact. TLI’s limitations are likely not to be communicated clearly. This is because predictive uncertainties can exacerbate patient anxiety during counseling [96].

Hence, the automation of embryo selection via TLI raises critical ethical dilemmas that necessitate careful management. This is more particularly in the key concern of psychological impact that touches on patient communication. Advancing beyond the theoretical concerns, it is crucial that clinics adopt proactive strategies to prevent misinformation and manage expectations effectively without harming patients in any significant way. The first concern ought to be mitigating algorithmic over-reliance and the “black-box” anxiety. For instance, as acknowledged earlier, patients and clinicians are likely to perceive TLI scores as definitive predictions, particularly in cases where there is tech overreliance. To counter this, communication should emphasize the advisory nature of the algorithm. For this, a more personalized communication approach would entail mainly framing TLI as a powerful decision-support tool rather than a definitive test. The appropriate analogy to use when interacting with patients would be comparing it to a sophisticated GPS. That is, it suggests the most probable route based on available data. However, it cannot guarantee traffic conditions or the final outcome of the journey. Just like an experienced driver, the embryologists should integrate this information with their expertise and the overall clinical picture. In practice, the conversation can be framed by one as *“The time-lapse system has given this embryo a favorable rating based on its development timing. This is one positive piece of information we consider alongside its appearance under the microscope.”*

The second concern is managing expectations and preventing high optimism in the technology. Naturally, the visual appeal of time-lapse videos can significantly inflate patient hopes. As such, clinics have a responsibility to contextualize this technology. The appropriate communication strategy would be to integrate a discussion of TLI’s limitations into the initial consent process. Specifically, technicians ought to mention the existing meta-data about the lack of universal improvement in live birth rates. In practice and when counseling the patient, a technician can state that: *“We use the time-lapse incubator because it provides a very stable environment for your embryos. It also gives us more information, specifically on developmental milestones. However, it is important to know that while this technology is advanced, current research shows that it does not guarantee a successful pregnancy. Its main benefit for us is in helping to refine our selection process, especially in complex cases.”*

The third and last significant concern is about transparency and predictive uncertainty. It is very critical that the probabilistic nature of predictions is clearly communicated to patients. This is due to the fact that it is essential for informed consent and psychological preparedness. Hence, an appropriate communication strategy would entail using numerical probabilities cautiously and always pair them with context. Further, technicians ought to avoid showing raw algorithm scores to patients without expert interpretation. In practice, an example touching on a lower-graded embryo that is the only one available for transfer would be: *“This embryo’s development was slightly slower than ideal according to the model. However, from experience, many babies have been born from embryos with similar patterns. The algorithm identifies statistical trends across large groups. What it cannot do is predict the potential of every single individual embryo.”*

### 4.6. Summary

TLI’s integration into IVF practice is hindered by several factors. The major ones are inconsistent evidence, technical limitations, and ethical complexities. Theoretically, TLI has significant advantages. However, its clinical value remains subject to standardizing protocols, validating algorithms across diverse settings, and mitigating costs. Instead of a universal, profit-oriented approach, a pragmatic, patient-centered adoption is crucial to aligning TLI’s use with evidence-based realities [14,51].

## 5. Morphokinetic Parameters and Predictive Algorithms

Morphokinetic markers are dynamic timestamps of embryonic development captured via TLI to predict viability. These include cleavage timings. For instance, t2, the time to 2-cell stage that occurs between 25 and 27 h post-insemination. Delayed t2, that is, more than 28 h, is indicative of reduced implantation potential and aneuploidy [47,91]. t3–t8 underlie the durations between subsequent cleavages. For instance, a shorter duration of the second cell cycle (t3-t2), typically under 12 h, reflects synchronized division and is associated with euploidy [77,91]. On the other hand, asynchronous cleavage such as t4–t3 greater than 1 h predicts blastocyst arrest [47]. There is also blastulation timing (tB), which refers to the time to blastocyst formation, typically between 96 and 120 h. Embryos that achieve tB of less than or equal to 116 h exhibit higher live birth rates [19,84]. On top of these are multinucleation and cytokinesis. Transient multinucleation at the 2-cell stage, which can only be detected by TLI, is linked to chromosomal abnormalities [21,97]. All have significant clinical relevance. First is aneuploidy prediction. Embryos with t5–t2 intervals greater than 30 h show 2.5-fold higher aneuploidy rates [91,98]. The second is blastocyst selection. Based on existing evidence, tB less than 110 h combined with even blastomere symmetry improves implantation rates by 18% in euploid embryos [84]. However, variability in culture conditions such as oxygen levels and media composition alters morphokinetic norms. This underlies the necessity of lab-specific benchmarks [27,62].

### 5.1. Commercial vs. Clinic-Specific Algorithms

The two major commercial algorithms covered in this article are Vitrolife and Merck. They are by no means the only ones. However, their leverage is that this analysis is premised on their popularity and significant adoption in the field. KIDScoreD5 (Vitrolife, Gothenburg, Sweden) operates by integrating t2, tB, and blastocyst expansion to rank embryos. In their 2022 study, Tartia et al. [83] reported 68% accuracy in predicting live birth in PGT-A cycles. On the other hand, Eeva (Merck - Darmstadt, Germany) leverages t2–t3 intervals and cleavage synchronicity. This is effective in homogeneous populations. However, as per available evaluative evidence, its accuracy drops in ethnically diverse patient populations due to differing morphokinetic baselines [42,97].

Regarding clinic-specific criteria, labs often develop in-house models tailored to local protocols. For example, Boucret et al. (2021) [18] created a clinic-specific algorithm prioritizing t5–t2 intervals. This achieved 22% higher pregnancy rates than commercial tools in recurrent implantation failure cases [18]. The key limitation with this is that clinic-specific models lack external validation. As such, they risk overfitting. Johansen et al. (2023) found that 60% of in-house algorithms failed generalizability tests across clinics [47].

From the above evidence, it is clear that commercial tools offer standardization. However, they struggle with population diversity [42,97]. In contrast, clinic-specific models provide customization. The caveat is that they require rigorous validation [47,66].

### 5.2. Role of AI and Machine Learning in Interpreting Time-Lapse Data

AI and machine learning (ML) are reshaping embryo selection significantly through automated morphokinetic analysis. In deep learning models, convolutional neural networks (CNNs) analyze time-lapse videos through which they predict ploidy and implantation potential. Highlighted in their paper, Lee et al., 2021 [52] developed an end-to-end AI model achieving 89% accuracy in ploidy prediction using t2–tB sequences. For hybrid AI systems, models that combine morphokinetic data with metabolomic or proteomic biomarkers such as IGF2 levels [21] improve predictive precision [30,43]. One of the advantages is reduced subjectivity. In this case, AI minimizes inter-embryologist variability. The outcome is standardized selection in multicenter settings [44,91].

The second advantage is hidden pattern detection. Machine learning identifies non-linear relationships such as t3–t5 intervals interacting with blastocyst morphology. These are attributes that humans overlook [97,98]. Regardless, challenges are also inevitable. One of them is dataset bias. For instance, AI trained on ethnicity is less useful for others. A case in point is that models based on Eurocentric datasets underperform in Asian or African populations [42,97]. The second issue is the black-box effect. Clinicians are less likely to trust AI decisions due to the unclear decision-making processes [44,91]. Last is the aspect of ethical risks. AI has its shortcomings in terms of objective discrimination. Embryos with atypical but viable morphokinetic profiles are likely to be discarded due to AI [66].

### 5.3. Summary

Morphokinetic markers and predictive algorithms underlie paradigm shifts in embryo selection. Their modus operandi is providing data-driven insights beyond static morphology. However, their clinical utility is constrained by several relevant factors. These include validation variability, population diversity, and ethical concerns. Overall, AI enhances objectivity. However, its integration requires transparent validation frameworks and multicultural datasets. The future lies in hybrid models that combine AI precision with embryologist expertise. This is on the premise that ethical and equitable application are guaranteed [44,91,97].

## 6. Time-Lapse Technology in Specific Contexts

### 6.1. Advanced Maternal Age (AMA)

TLI’s ability to detect subtle discrepancies in key morphokinetic markers is particularly relevant for AMA patients (≥38 years). This is because they face higher rates of aneuploidy and embryo arrest. Several morphokinetic advantages are evident. For instance, embryos from AMA patients often exhibit delayed cleavage t2 that is greater than 28 h and prolonged blastulation of tB greater than 120 h [23,95]. These markers are linked to aneuploidy [23,95]. TLI excels in identifying these anomalies. As a result, it enables the prioritization of embryos with synchronized divisions such as t3–t2 of less than 10 h, which is an attribute linked to euploidy in AMA cohorts [73,95]. Further, in their review, Sainte-Rose et al., 2021 [73] report a 15% increase in blastocyst formation rates in AMA patients using TLI. The outcome was attributed to avoiding static evaluation-induced stress [73]. Based on these strengths, the clinical outcomes are significant. For instance, a retrospective study by Chen et al., 2018 [23] found TLI improved implantation rates by 12% in AMA patients compared to static methods. This supports the notion that TLI may aid in selecting the most viable embryos from a cohort with higher aneuploidy risk. However, these benefits are not universal (see Section 4.1). This variance highlights the impact of clinic-specific protocols and patient heterogeneity. Furthermore, a critical limitation is that morphokinetic norms for AMA embryos vary widely. This necessitates adjusting clinic-specific factors to avoid discarding viable embryos with atypical timelines [10,95].

### 6.2. Recurrent Implantation Failure (RIF)

In RIF patients, TLI excels in identifying embryos with hidden viability. This tends to be missed by conventional grading. Based on mechanistic insights, RIF embryos often show transient multinucleation or irregular cleavage synchronicity. This is only detectable via TLI [28,57]. For instance, Rubio et al., 2014 [71] observed a 23% higher pregnancy rate in RIF patients using TLI. The outcome is because anomalies like direct cleavage were excluded [71]. In another study, Kozyra et al. (2024) [50] developed a TLI-based algorithm for RIF. They prioritized t5–t2 intervals of less than 30 h and the outcome was an improved live birth rates by 18% [50]. Regarding clinical outcomes, a 2024 RCT by Stevens Brentjens et al. [80] reported a 27% ongoing pregnancy rate in RIF patients using TLI versus 19% with static methods. However, TILT trial subgroup analyses found no significant benefit for RIF. This highlights protocol-dependent variability [14].

TLI synergizes with PGT-A by preselecting embryos likely to be euploid. The evident advantages are reduced biopsy costs and lab workload. On the aspect of enhanced selection efficiency, TLI parameters such as tB of less than 116 h correlate strongly with euploidy. In their study, Popovic et al., 2024 [66] noted that TLI preselection reduced PGT-A biopsy cycles by 32% without compromising euploid yield. In another study, Watanabe et al. (2024) [92] demonstrated that embryos with ideal morphokinetics such as t2–t5 intervals of less than 25 h had 89% concordance with PGT-A results. The result is that unnecessary biopsies are minimized [92]. Regarding pertinent clinical outcomes in PGT-A cycles, TLI improves euploid embryo identification by 20%. However, as per available evidence, the live birth rates remain comparable to morphology-based selection [8,82]. Further cost–benefit analyses suggest TLI + PGT-A is most viable for clinics with high patient volumes [82].

### 6.3. Comparisons of Outcomes with Traditional Morphology Evaluation

When comparing TLI outcomes to traditional morphology evaluations, evidence suggests that TLI is superior in its predictive capabilities when it comes to establishing embryo viability. Traditional methods often rely on static assessments at specific time points. The downside is that the approach tends to miss critical developmental milestones [8,28]. In contrast, TLI captures continuous data. This allows for a comprehensive analysis of embryonic development [50,57]. Numerous studies have demonstrated that TLI leads to higher implantation and live birth rates compared to conventional methods. The difference is significantly evident in complex cases such as advanced maternal age and RIF [64,71,92]. Table 4 below presents a summative preview of the comparisons.

### 6.4. Summary

TLI demonstrates niche superiority in AMA, RIF, and PGT-A contexts. This is because the dynamic markers enhance embryo selection precision. However, its benefits over traditional morphology are not universal as per existing evidence. They rely heavily on patient stratification and clinic protocols. TLI reduces subjectivity and improves communication. However, its high costs and variable validation limit broad adoption. As such, hybrid models, that is, those combining TLI’s granularity with morphology’s practicality, may optimize outcomes in targeted populations [57,82,95].

## 7. Future Directions and Recommendations

### 7.1. Personalized Embryo Selection Protocols

Current one-size-fits-all TLI algorithms lack precision for heterogeneous patient populations. As such, future protocols should aim to integrate patient-specific factors such as age, ovarian reserve, and genetic predispositions with morphokinetic data. The first significant issue is AI-driven personalization. Deep learning models trained on multi-modal datasets can be efficient in yielding superior outcomes. For instance, models trained on morphokinetics and patient genomics could predict individualized embryo viability. For instance, in their study, AlSaad et al. (2025) [4] developed a model that adjusts tB (blastulation timing) thresholds based on maternal age and anti-Müllerian hormone levels. They improved live birth rates by 14% in poor responders [4]. The second relevant aspect is dynamic risk stratification. Algorithms could prioritize embryos with resilient morphokinetic profiles such as stable t3–t2 intervals despite oxidative stress. This is needed, particularly for patients with endometriosis or obesity [77,91]. A sufficient recommendation is that stakeholders need to develop validated, as well as adjustable algorithms that account for patient biomarkers and clinical history. This is necessary to move beyond static morphokinetic thresholds [4,85].

### 7.2. Integration with Non-Invasive Biomarkers

Combining TLI with omics-based biomarkers such as metabolomics and proteomics may enhance predictive accuracy. At the same time embryo integrity will be preserved. One way is through Spent Culture Media (SCM) Analysis. SCM has metabolites such as glutamate and Soluble Human Leukocyte Antigen-G (sHLA-G). These correlate with embryo viability. TLI timestamps such as t5 can guide sampling windows, thus optimizing biomarker detection [35,63]. For instance, Ghelardi 2022 [35] proposed pairing tB with SCM lactate levels to predict euploidy with 82% accuracy. The second approach is through exosome profiling. Nanoparticle tracking of embryo-derived exosomes in culture media needs to be timed to morphokinetic milestones such as compaction. There is a high chance that it may reveal non-invasive ploidy markers [85]. Based on these, it is thus recommended that developers standardize protocols for concurrent TLI and biomarker analysis. This is to foster the synchronization of data capture [63,85].

### 7.3. Data Standardization Across Centers

The lack of universal morphokinetic benchmarks undermines TLI’s reliability. Firstly, there is culture condition variability. Differences in oxygen tension (5% vs. 20%), media composition, and incubation protocols alter t2–tB intervals by up to 6 h. This impact is skewed algorithm predictions [68,76]. There is also the issue of reporting frameworks. Less than 30% of clinics adhere to consensus guidelines such as the European Society of Human Reproduction and Embryology (ESHRE) TLI reporting standards. This leads to inconsistent data collection [69,89]. Key recommendations include the fact that actors need to adopt universal culture protocols such as fixed oxygen levels and standardized media for TLI studies [68]. Secondly, mandatory reporting of key variables such as lab temperature and pH alongside morphokinetic data needs to be implemented [89].

### 7.4. Multicenter Collaboration for Model Training

AI algorithms trained on single-center data are likely to suffer from poor generalizability as confirmed by existing evidence. Potential solutions to the generalizability dilemma include shared databanks. Platforms like the Global TLI Consortium as proposed by Wang et al., 2023 [89,90], could aid the pooling of de-identified morphokinetic data, biomarkers, and outcomes from diverse populations. Another key solution is federated learning. AI training needs to be decentralized. Models that learn from multi-center data without transferring sensitive records are likely to reduce bias [13]. Lee et al., 2021 [52] demonstrated a 22% improvement in ploidy prediction accuracy using federated learning across Asian and European clinics. A recommendation based on these is that stakeholders and key actors need to fund international consortia to create open-access TLI datasets. They also need to validate algorithms across ethnicities and lab conditions [4,52,78].

### 7.5. Synthesis and Strategic Roadmap

To realize TLI’s full potential, the field must first prioritize personalization. The transition from population-based to patient-specific embryo selection using AI and integrated biomarkers is necessary to guarantee operational efficiency [85,91]. The second key aspect is standardization. Regulatory authorities and other key actors ought to establish global guidelines for TLI data collection, as well as reporting to reduce inter-lab variability [68,89]. The third aspect is collaboration. Key stakeholders and relevant interested parties need to build multicenter partnerships to train AI models that are robust and generalizable [78,90]. The last aspect is ethical oversight. Regulators need to ensure transparency in algorithm design. Further, the overmedicalization of embryo selection needs to be avoided [76].

## 8. Conclusions

Based on the extensive evidence reviewed, TLI represents a significant advancement in IVF. It offers continuous, non-invasive embryo monitoring and data-driven morphokinetic analysis. One of the key benefits highlighted is enhanced Embryo Selection. TLI excels at identifying dynamic markers such as t2, tB linked to viability. This improves selection accuracy in niche populations like RIF and AMA [23,50,71]. Secondly, TLI reduces subjectivity. Through its automated algorithms, it mitigates embryologist bias, thus standardizing evaluations across clinics [38,44]. Thirdly, it fosters patient engagement. Unlike the traditional approach, TLI has visual timelines that ensure transparency and informed decision-making [6,65].

Nevertheless, despite these highlighted efficiencies, clinical evidence remains mixed. Specifically, the central finding from large-scale trials is TLI’s lack of universal superiority in improving live birth rates. Its primary value appears to be in niche applications, which are contingent upon standardized protocols and validated algorithms. Further, compared to conventional evaluation, TLI exhibits substantial cost and ethical concerns. High equipment costs, alongside the risks of overreliance on unvalidated predictive models, are likely to hinder accessibility and trust [5,51,61].

### 8.1. Re-Emphasizing Critical Adoption over Blind Enthusiasm

The integration of TLI demands a measured, evidence-based approach for efficiency. Firstly, TLI applications need to be targeted. This means prioritizing TLI in subgroups such as RIF and PGT-A cycles where morphokinetic precision adds value, rather than universal adoption [50,81,82]. Secondly, TLI needs to be standardized and validated. Universal protocols for culture conditions, data reporting, and algorithm training to reduce variability and bias need to be established by regulatory authorities and key stakeholders [68,89]. Thirdly, TLI adoption needs substantial ethical vigilance. More particularly, algorithmic determinism should be avoided. Further, clinicians need to maintain oversight to balance AI insights with clinical judgment [25,51]. Fourthly, a collaborative initiative is needed, particularly in innovation. Stakeholders need to invest in multicenter consortia to pool diverse datasets and refine AI models. This will ensure equitable applicability [78,90]. Fifthly, the development of standardized patient communication guidelines for TLI is urgently needed. Professional societies such as ESHRE and the American Society for Reproductive Medicine should lead initiatives to create consensus statements and training materials. These need to guide professionals on how to responsibly disclose the capabilities and limitations of predictive algorithms, in the process ensuring equitable and informed patient consent across clinics.

### 8.2. Final Perspective

TLI is not a magic formula. Rather, it is a powerful adjunct in the IVF toolkit. Contrary to existing beliefs, TLI’s true potential lies not in displacing conventional methods but in complementing them through strategic, evidence-guided adoption. By addressing current limitations through undertakings such as fostering standardization, validation, and ethical stewardship, the field can harness TLI’s strengths while mitigating its risks. This will ultimately advance personalized and equitable reproductive care for a larger majority globally.

### 8.3. Clinical Practice Points

Based on the current evidence reviewed, the following conclusions can be drawn for clinical practice:Targeted Application, Not Universal Adoption: The routine use of TLI for all IVF patients is not justified by current evidence. However, its value is most pronounced in specific clinical scenarios:
○PGT-A cycles: TLI can act as a pre-screening tool to prioritize embryos for biopsy. This has the potential to reduce laboratory workload and costs.○RIF and AMA: In these populations, TLI has the ability to detect dynamic anomalies such as abnormal cleavages. This may improve embryo selection precision over static morphology alone.No Universal Superiority in Live Birth Rates: Existing evidence from large trials shows that TLI does not significantly improve live birth rates compared to conventional incubation and morphological assessment.Cost–Benefit Considerations are Paramount: TLI systems have high upfront and operational costs. Hence, a cost–benefit analysis is essential. This is because the marginal gains in selection accuracy rarely justify the expense for average-prognosis patients.Algorithmic Insights are Advisory, Not Deterministic: The predictive algorithms used in TLI are tools to aid embryologist judgment; they are not there to replace it. Clinicians must maintain oversight and communicate the probabilistic nature of these predictions to patients. This is necessary to manage expectations appropriately.Standardization is a Prerequisite for Validity: The clinical utility of TLI is highly dependent on the core aspects of standardized laboratory protocols and validated, context-specific algorithms. The current widespread variability in practice limits the generalizability of findings and tool performance.

## Figures and Tables

**Figure 1 ijms-26-09609-f001:**
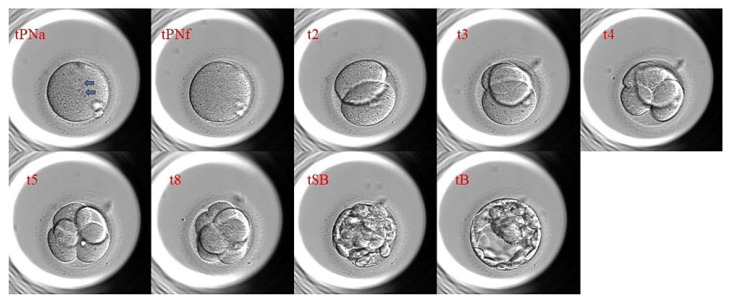
Adopted from Taniguchi et al., *Cureas* 2024 under CC BY 4.0. It highlights key morphokinetic parameters that are leveraged in embryo selection [79].

**Table 1 ijms-26-09609-t001:** Comparative Summary of TLI Incubators (Embryoscope and Eeva).

Feature	EmbryoScope (Vitrolife)	Early Embryo Viability Assessment—Eeva (Merck KGaA)
**Core Technology**	Integrated microscope and camera Continuous imaging in a stable incubator.	Automated algorithms for early-stage morphokinetic analysis (first 48 h).
**Primary Strength**	Comprehensive morphokinetic profiling (t2 to tB) Widely adopted and studied Improves workflow standardization.	Aims to simplify and standardize selection. This is potentially beneficial for labs with less embryology expertise.
**Key Algorithm/Focus**	Proprietary algorithms, mainly EmbryoScope+, which uses a wide range of morphokinetic parameters.	Generates a viability score based on early cleavage events. For instance, first cytokinesis.
**Main Criticism/Limitation**	Questionable clinical superiority. This is because it fails to consistently and significantly improve live birth rates over conventional methods [2]. It is also expensive	Limited predictive scope. The reliance on early-stage markers raises questions about accuracy for blastocyst-stage outcomes [31].
**Common Challenges**	Lack of generalizability. The performance of algorithms varies across different labs and patient populations [7,33]. High cost of equipment and required expertise hinder widespread adoption [8,15].	High cost-to benefit ratio. Limited algorithm generalizability. Limited predictive window. Context-dependent benefits.

**Table 2 ijms-26-09609-t002:** Summarizes the key differences between continuous monitoring and static evaluation.

Feature	Continuous Monitoring (TLI)	Static Evaluation	Key References
**Data Capture**	Dynamic, uninterrupted morphokinetic tracking	Snapshot assessments at fixed intervals such as Day 3 or Day 5	[49,51,59]
**Embryo Stress**	Minimal—embryos remain in stable culture conditions	High—repeated removal from incubator	[15,16,17,18,19]
**Detection of Anomalies**	High—Identifies transient events (multinucleation, abnormal cleavages)	Low—misses dynamic anomalies	[7,49]
**Subjectivity**	Low—algorithm-driven analysis	High—depends on the expertise of the embryologist	[51,59]
**Aneuploidy Correlation**	Stronger link via morphokinetic markers (e.g., delayed t2/tB)	Weak correlation with ploidy	[51]
**Cost and Accessibility**	High—expensive equipment and training needed	Low—widely accessible	[5,61]
**Clinical Utility**	Context-dependent—superior in RIF/AMA; mixed in general IVF	Consistent but limited in complex cases	[14,23,50]

**Table 3 ijms-26-09609-t003:** Summary of the proposed promises of Time-Lapse Technology, supporting evidence, and key limitations.

Promise of TLI	Proposed Mechanism and Supporting Evidence	Limitations and Contradictory Evidence
**Improved Embryo Selection Accuracy**	Continuous monitoring captures dynamic morphokinetic parameters such as t2, tB, and cleavage synchronicity amongst others. It also captures transient anomalies that tend to be missed by static assessment such as multinucleation and direct cleavage [49,59,67]. TLI also links certain specific patterns such as delayed t2/tB to aneuploidy and reduced viability [11,75,91].	The generalizability of algorithms is limited by two key factors: inter-clinic variability in culture conditions and diversity of patient populations [47,51]. Large RCTs like SelecTIMO [48] and TILT [14] have found no significant improvement in overall live birth rates versus conventional methods.
**Reduced Embryologist Subjectivity**	Automated, algorithm-driven analysis standardizes embryo evaluation. This in turn reduces inter-observer variability. For instance, Armstrong et al., 2022 [6] reported a 30% reduction in grading discrepancies. On top of this, AI integration further minimizes human bias [38,44].	Significant risk of overreliance on yet-to-be-validated algorithms [25,51]. It also requires significant training and expertise to interpret data correctly. This will not completely phase out subjectivity but shift its nature [25].
**Increased Implantation and Live Birth Rates (LBR)**	Observational studies and early single-center trials reported 15–20% higher implantation rates. This is more particularly in niche populations like RIF [71] and AMA [23].	The major multicenter TILT RCT Bhide et al., 2024 [14] found no significant difference in LBR (32.1% vs. 31.4%). SelecTIMO RCT Kieslinger et al., 2023 [48] also reported comparable LBRs. Overall, pertinent benefits appear highly context-dependent.
**Enhanced Patient Communication and Transparency**	Visual timelines of embryo development improve patient understanding of embryo quality. It also helps with their grasp of the treatment rationale. On this, studies report higher patient satisfaction and the feeling of being informed among individuals [6,65].	Presents ethical concerns due to predictive algorithms being communicated as overly definitive. This potentially inflates patient expectations and anxiety if outcomes are either unsuccessful or what they did not expect [51].
**Superiority in Specific Populations: AMA**	Detects subtle morphokinetic delays (t2 > 28 h, tB > 120 h). These are linked to higher aneuploidy rates in AMA patients. In their report, Chen et al., 2018 [23] note a 12% improvement in implantation rates compared to static methods in AMA.	Chera-Aree et al. (2021) [24] found no significant difference in pregnancy outcomes between TLI and conventional incubation. This was after an age-stratified analysis. Live birth rates have also often remained comparable despite improved implantation [23].
**Superiority in Specific Populations: RIF**	TLI identifies dynamic dysmorphisms, that is, irregular cleavages, which are the main cause of previous failures. In their study, Rubio et al. (2014) observed a 23% higher pregnancy rate in RIF patients using TLI [71]. Stevens Brentjens et al., 2024 [80] also reported a 27% ongoing pregnancy rate compared to 19% with static methods.	Subgroup analysis of the large TILT trial [14] found no significant benefit for RIF patients. This highlights the fact that the positive outcomes may be protocol-dependent, thus not universal.
**Synergy with PGT-A**	Acts as a pre-screening tool to preselect embryos likely to be euploid based on morphokinetics. This significantly reduces the number of unnecessary biopsies. For instance, Popovic et al., 2024 [66] noted a 32% reduction in biopsy cycles.	While improving euploid identification efficiency, live birth rates often remain comparable to morphology-based selection in PGT-A cycles [8,82]. This undermines and questions TLI’s additive value in all cases [8,82].

**Table 4 ijms-26-09609-t004:** Comparison of TLI outcomes and traditional morphology outcomes.

Metric	TLI Outcomes	Traditional Morphology Outcomes	Key Studies
Embryo Selection	Prioritizes dynamic markers (t2, tB); reduces subjectivity	Rely on static snapshots (cell number, fragmentation)	[57,71,73]
Aneuploidy Detection	65–75% accuracy via morphokinetics vs. 50–60% with morphology	Limited to indirect markers (fragmentation, asymmetry)	[23,82,94]
Live Birth Rates	Context-dependent: +15–20% in RIF/AMA; no difference in general populations	Consistent across broad populations but lower in complex cases	[14,50,80]
Cost Efficiency	High upfront costs; justified in PGT-A/RIF cohorts	Lower costs but higher repeat cycles in complex cases	[8,61,82]

## Data Availability

No new data were created or analyzed in this study. Data sharing is not applicable to this article.

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
