# Peer review of "Time-Lapse Imaging in IVF: Bridging the Gap Between Promises and Clinical Realities"

_ijms, 2025, doi:10.3390/ijms26199609_

Round 1
Reviewer 1 Report
Comments and Suggestions for Authors
1. Brief Overview
This review article discusses the growing popularity of incubators with time-lapse imaging systems in human IVF, which allow continuous monitoring of embryonic development without the need to remove embryos from the incubator. The initial promise of this technology was to improve embryo selection, optimize IVF outcomes, and deepen the understanding of human embryology. The authors discuss the main promises and potential benefits, such as better embryo selection, reduced embryonic stress, improved understanding of embryology, and communication with patients. The authors also discuss some clinical realities and challenges of this technology, such as inconsistent evidence, costs and learning curve, algorithm validation, and the need for artificial intelligence tools. Finally, the article concludes that although time-lapse technology has enormous potential for research and optimization of the embryonic environment, and has become a valuable tool in the clinical practice of many IVF clinics, it has not yet fully realized its potential to revolutionize embryo selection and drastically improve clinical outcomes.
2. General comments on the concept
It is a excellent article. The enhancement of time-lapse imaging in human in vitro fertilization will significantly contribute to increasing birth rates.
3. Specific comments
It would be very important to discuss more about time-lapse incubators, as only two technologies were mentioned, while other brands offer more advanced technologies. For example, the GERI incubator differs from those mentioned in the article, as it consists of multiple incubation cameras with an independent microscope and image capture system.
Author Response
Comment 2: "It is an excellent article. The enhancement of time-lapse imaging in human in vitro fertilization will significantly contribute to increasing birth rates."
Response: We are sincerely grateful for your positive remarks. This is encouraging for us to continue pursuing this subject further.
Comment 3: "It would be very important to discuss more about time-lapse incubators, as only two technologies were mentioned, while other brands offer more advanced technologies. For example, the GERI incubator differs from those mentioned in the article, as it consists of multiple incubation cameras with an independent microscope and image capture system."
Response: Thank you for suggesting this. We indeed agree with your proposition that more TLI technologies ought to be discussed, as we only covered two. Alongside GERI, we also found MIRI TL. This can bring the total to 4 technologies. However, as a minor revision, how do we go about it so that it does not involve making major changes? Discussing these platforms means introducing new citations, which will have to be fitted in a chronological order. A mere citation introduction means that all subsequent in-text numbers have to change. Kindly advise accordingly, as we believe the additions would strengthen the manuscript further, but it is also still okay in its current form without the additions (a stepping stone or precursor of future reviews)
Reviewer 2 Report
Comments and Suggestions for Authors
In this article “Time-Lapse Imaging in IVF: Bridging the Gap between Promises and Clinical Realities.” Authors have done a literature review about the Time-lapse imaging (TLI) which has emerged as a transformative technology in in vitro fertilization (IVF) as it offers continuous, non-invasive embryo assessment through morphokinetic profiling. Authors discuss its key advantages such as reduced embryologist subjectivity, detection of dynamic anomalies, and improved implantation rates in niche populations. However, its clinical utility remains debated. Large trials and meta-analyses reveal no universal improvement in live birth rates compared to conventional methods. Authors also discuss Key challenges underlying the outcome include algorithm generalizability, lab-specific protocol variability, and high costs. Nevertheless, TLI shows promise in specific contexts. For instance, PGT-A cycles where it reduces unnecessary biopsies by predicting euploidy. However, even in this, its benefits are marginal in unselected populations. This review synthesizes evidence to highlight that TLI’s value hinges on standardization, rigorous validation, and tailored application. As such, adoption must be cautious to avoid resource misallocation without significant IVF outcome improvements. In future, personalized protocols, integration with non-invasive biomarkers, and multicenter collaboration are crucial to optimize TLI’s potential in assisted reproduction. However, authors have not given extensive data about the subject and should include following in the revised manuscript for readers to understand and efficiently use the information in their research questions.
Please find my comments about the manuscript.
- Please use full form first in the text and then use short forms to understand well. For example, when using for the first time terms like (PGT-A), sHLA-G and so son.
- Authors have included data in table 1 Summarizing the key differences between continuous monitoring and static evaluation. Similarly, authors MUST include a table mentioning all the Promises of Time-Lapse Technology that have been discussed in the review and add more details about trials which positively and negatively impact AMA and RIF which will give a quick recap about everything in the review for the ease and increase of readership.
- Authors in general have mentioned a very few studies about the involvement of TLI in IVF otcomes. Authors should be citing more literature about the evidence of TLI success/failure in revised review.
Please include all the changes in the revised manuscript.

Author Response
Comment 1: Please use full form first in the text and then use short forms to understand well. For example, when using for the first time terms like (PGT-A), sHLA-G and so son
Response: Thank you for point this out, considering that it is crucial for improved readability. We have made the relevant changes, specifically section 2.3 where most terms had not been defined
Comment 2: Authors have included data in table 1 Summarizing the key differences between continuous monitoring and static evaluation. Similarly, authors MUST include a table mentioning all the Promises of Time-Lapse Technology that have been discussed in the review and add more details about trials which positively and negatively impact AMA and RIF which will give a quick recap about everything in the review for the ease and increase of readership.
Response: We agree with your suggestion. We have now included a new Table 2 (at the end of section 3) that synthesizes the key promises of Time-Lapse Technology, the supporting evidence, and the associated limitations or contradictory findings, as discussed throughout the manuscript. This table provides a concise overview for the reader.
We also agree that a more detailed synthesis of the evidence for these key subgroups is beneficial. This was already covered in the narrative within Sections 6.1 and 6.2. They explicitly name and discuss the pivotal trials and studies that have reported both positive and negative outcomes for these populations.
Comment 3: Authors in general have mentioned a very few studies about the involvement of TLI in IVF otcomes. Authors should be citing more literature about the evidence of TLI success/failure in revised review.
Response: We do believe we have provided enough evidence, given the existing pool of literature. Not many key studies exist. However, we have tried our best to cite the significant ones. On top of them, we have provided other supporting pieces of literature, which can be confirmed by our citations in the various sections, specifically section 3

Reviewer 3 Report
Comments and Suggestions for Authors
Review for Time-Lapse Imaging in IVF: Bridging the Gap between Promises and Clinical Realities
The manuscript provides a comprehensive and well-structured summary of the place of time-lapse imaging (TLI) in in vitro fertilization. The authors structure the article logically, starting with the technological basics, moving on to clinical promises and limitations, and ending with future directions. A strength of the manuscript is that it presents the advantages and weaknesses of TLI in a balanced manner, not in a promotional but in a critical way. The reference list is extensive, includes several recent references, and the authors are sensitive to the contradictions in the clinical evidence.
At the same time, the text in its current form is too long and redundant in several places. Several arguments, such as the validation shortcomings of predictive algorithms or the limited impact of TLI on live birth rates, are repeated several times in different chapters. To improve readability, it is recommended that these be condensed. Similarly, the description of the technology (especially the EmbryoScope and Eeva systems) is disproportionately detailed and reads more like a manufacturer's brochure than a critical review. It would be more appropriate to present this information in a shorter, comparative table or figure.
The section on ethics and patient communication is currently shorter and less nuanced than the section on technology. It would be useful to provide more specific examples of how to communicate responsibly to patients about the inaccuracy of algorithms and the uncertainty of predictions, and how to prevent the development of excessive expectations. As an active EmbryoScope user, I would find it particularly interesting and useful if the authors supplemented the manuscript with specific communication strategies and practical examples.
The conclusion of the manuscript is balanced, but it lacks a clear take-home message. A short summary box or bullet-point message with the most important clinical conclusions (e.g., that TLI is mainly useful in RIF and AMA cases and in PGT-A cycles, that its universal application is not justified, and that high costs limit its widespread introduction) would be very helpful to the reader.
Overall, the manuscript is a timely and useful review that represents an important contribution to the literature. I consider it suitable for publication, but it could be significantly improved by addressing the minor shortcomings mentioned above (condensing, strengthening take-home messages). Accordingly, a minor revision is recommended.
Author Response
Comment 1: At the same time, the text in its current form is too long and redundant in several places. Several arguments, such as the validation shortcomings of predictive algorithms or the limited impact of TLI on live birth rates, are repeated several times in different chapters. To improve readability, it is recommended that these be condensed.
Response: After reviewing, we agree with this suggestion. we have condensed the arguments in other sections, leaving the master argument for sections 4.1 for live birth rate, 4.2 for algorithm validation, and 4.4 for cost. We have now only highlighted the argument in other areas, including the abstract and conclusion, being keen not to give away everything as we had initially done
Comment 2: Similarly, the description of the technology (especially the EmbryoScope and Eeva systems) is disproportionately detailed and reads more like a manufacturer's brochure than a critical review. It would be more appropriate to present this information in a shorter, comparative table or figure.
Response: We have tried to reduce the description, specifically for embryoscope, and provided a comparative summary for the two systems (Table 1 - section 2.1)
Comment 3: The section on ethics and patient communication is currently shorter and less nuanced than the section on technology. It would be useful to provide more specific examples of how to communicate responsibly to patients about the inaccuracy of algorithms and the uncertainty of predictions, and how to prevent the development of excessive expectations. As an active EmbryoScope user, I would find it particularly interesting and useful if the authors supplemented the manuscript with specific communication strategies and practical examples.
Response: We have addressed this concern by significantly expanding section 4.5 where we integrated specific, practical communication strategies and example phrases directly into the text. We also added a paragraph to section 3.4, balancing the promised benefits with the necessary cautions. In section 8.1, we added a fifth recommendation about the need to develop standardized patient communication guidelines
Comment 4: The conclusion of the manuscript is balanced, but it lacks a clear take-home message. A short summary box or bullet-point message with the most important clinical conclusions (e.g., that TLI is mainly useful in RIF and AMA cases and in PGT-A cycles, that its universal application is not justified, and that high costs limit its widespread introduction) would be very helpful to the reader.
Response: We have added a new final segment (8.3 - clinical practice points), which perfectly summarizes the most important clinical conclusions in a bullet-point format
